# Exercise, Comorbidities, and Health-Related Quality of Life in People Living with HIV: The HIBES Cohort Study

**DOI:** 10.3390/ijerph17145138

**Published:** 2020-07-16

**Authors:** Philipp Zech, Felipe Schuch, Camilo Pérez-Chaparro, Maria Kangas, Michael Rapp, Andreas Heissel

**Affiliations:** 1Social and Preventive Medicine, Department of Exercise and Health Sciences, University of Potsdam, 14469 Potsdam, Germany; 2Department of Sports Methods and Techniques, Federal University of Santa Maria, 97105 Santa Maria, Brazil; felipe.schuch@ufsm.br; 3Outpatient Clinic—Center for Sports Medicine, Department of Sports & Health Sciences, University of Potsdam, 14469 Potsdam, Germany; perezcha@uni-potsdam.de; 4Department of Psychology, Centre for Emotional Health, Macquarie University, 2109 Sydney, Australia; maria.kangas@mq.edu.au; 5Social and Preventive Medicine, Department of Sports and Health Sciences, Intra-faculty unit “Cognitive Sciences”, Faculty of Human Science, and Faculty of Health Sciences Brandenburg, Research Area Services Research and e-Health, University of Potsdam, 14469 Potsdam, Germany; mrapp@uni-potsdam.de (M.R.); andreas.heissel@uni-potsdam.de (A.H.)

**Keywords:** HIV, exercise intensity, quality of life, comorbidity

## Abstract

*(1) Background*: People with HIV (PWH) may perform more than one type of exercise cumulatively. The objective of this study is to investigate recreational exercise and its association with health-related quality of life (HRQOL) and comorbidities in relation to potential covariates. *(2) Methods:* The HIBES study (HIV-Begleiterkrankungen-Sport) is a cross-sectional study for people with HIV. The differences between non-exercisers versus exercisers (cumulated vs. single type of exercises) were investigated using regression models based on 454 participants. *(3) Results:* Exercisers showed a higher HRQOL score compared to non-exercisers (Wilcox r = 0.2 to 0.239). Psychological disorders were identified as the main covariate. Participants performing exercise cumulatively showed higher scores in duration, frequency, and intensity when compared to participants performing only one type of exercise. The mental health summary score was higher for the cumulated and single type of exercise if a psychological disorder existed. Duration and intensity were associated with an increase of HRQOL, whilst a stronger association between psychological disorders and exercise variables were evident. Exercise duration (minutes) showed a significant effect on QOL (standardized beta = 0.1) and for participants with psychological disorders (standardized beta = 0.3), respectively. *(4) Conclusions:* Psychological disorders and other covariates have a prominent effect on HRQOL and its association with exercise. For PWH with a psychological disorder, a stronger relationship between HRQOL with exercise duration and intensity emerged. However, differentiation of high-HRQOL individuals warrants further investigation by considering additional factors.

## 1. Introduction

Due to antiretroviral therapy (ART), human immune deficiency virus (HIV) has evolved into a chronic disease [1]. Epidemiological data for the number of people living with HIV (PWH) in Germany for 2018 was *n* = 87,900 cases [2]. The odds of low health related PWH can be compared with the odds of other chronic diseases [3]. When compared to the population without HIV, PWH, even if immunologically and virologically stable, present with significantly lower health related quality of life (HRQOL) in several domains [4].

The prevalence of multimorbidity is significantly higher in PWH compared to HIV-negative persons (63% vs. 43%, *p* < 0.001). The duration of HIV and antiretroviral drugs are also associated with multimorbidity in PWH [5]. A retrospective analysis for a German HIV population found similar results. The prevalence of acute renal disease, Hepatitis C, bone fractures, and cardiovascular disease were significantly higher in PWH compared to HIV negative persons [6]. The study by Gallant et al. found that comorbidities are common among an aging HIV population and that comorbidities, especially Hypertension and Hyperlipidemia, increased over time [7].

The benefits of different types of physical exercise such as aerobics, strength training, yoga, tai chi, and other forms of exercise for PWH are well known [8,9,10,11,12,13,14]. However, recent literature has focused on exploring the role of intensity and total volume of physical activity [15,16,17], but limited attention has been given to whether recreational exercises, other than aerobics, strength training, yoga, or tai chi itself, have specific effects on HRQOL. Furthermore, HRQOL is associated with comorbidities in an HIV positive sample [18,19].

A current research gap is that some people may perform more than one type of recreational exercise cumulatively on a weekly basis, such as resistance training, soccer, and tennis; yet this has yet to be empirically examined. A further research gap is that exercise frequencies, volume, and intensity and their effects on comorbidities and HRQOL are scarcely investigated. This everyday life experience and practice needs to be taken into account when investigating PWH in regard to HRQOL and comorbidities. A multitude of benefits are associated with multiple exercise participation, including changing demands on body and mind, muscle activation in diverse ways, avoiding overuse of tissues, improving coordination, and preventing burn-out [20,21]. Performing more than one type of exercise can also result in greater exercise volume. However, neither the prevalence of cumulated types of exercise (CTE) that PWH perform, nor its frequency, duration, and intensity has been investigated in relation to HRQOL and present comorbidities.

To date, no published study has investigated recreational exercise (independent outcome) and its relation with HRQOL parameters (primary dependent outcome) with regard to comorbidities (secondary outcome). The following study addresses the differences between single types of exercise (STE) and CTE. CTE may serve as a mediating intervention to improve HRQOL for PWH with or without a comorbidity. The objective of this study was twofold. The first objective was to investigate differences in exercising (STE vs. CTE) and non-exercising participants with HIV in relation to recreational exercise training variables (specifically, frequency, volume, and intensity). The second objective was to evaluate these exercise training variables in association with psychological and cardio-metabolic comorbidities, and HRQOL in PWH. Three hypotheses were tested: (i) exercise is associated with better HRQOL, (ii) Greater exercise frequency, volume, and intensity is expected to be positively related to HRQOL, (iii) Exercising PWH with a comorbidity have greater benefits for their HRQOL than exercising PWH without a comorbidity.

## 2. Materials and Methods

A cross-sectional design was used for this study. The HIBES (HIV, Begleiterkrankungen, Sport, translation: HIV, Comorbidities, exercise) questionnaire included data on socio-demographics, HIV and anti-retroviral medication, psychological and cardio-metabolic comorbidities and HIV-specific or Acquired Immune Deficiency Syndrome (AIDS)-defining illnesses, recreational exercise and its training volume (exercise sessions and minutes), and HRQOL. The study was conducted from October 2010 to December 2012 over a 26-month period in both rural and urban areas of Germany.

The HIBES cohort study was reviewed and approved by the Ethics Committee of the Charité Berlin (Protocol No. EA1/084/11) and is in accordance with the Declaration of Helsinki. Written informed consent was granted. Accordance with data protection and anonymity of the HIBES study was declared by the data protection officer of the Humboldt University of Berlin.

### 2.1. Inclusion Criteria

Eligible individuals were required to be ≥18 years of age, diagnosed positive with HIV, and completed questionnaires on exercise and HRQOL.

### 2.2. Participant and Public Involvement

The study aims and design were developed by sport scientists, professionals for HIV and AIDS, physicians, and public institutions for HIV in Germany. Participants were not involved in the study design, conduct, or dissemination plan of the research, nor were they invited to contribute to the writing or editing of this document for readability or accuracy.

### 2.3. Recruitment

In order to minimize the recruitment bias, participants were recruited from a wide range of institutions that are involved in AIDS care or media. Participants were recruited from (1) The official AIDS-offices (Germany-wide), (2) the Academy Waldschlösschen e.V., (https://www.waldschloesschen.org/de/), (3) Medical care facilities in Berlin and Germany who specialize in HIV and AIDS, (4) The consortium of HIV and AIDS specialized physicians (DAGNÄ) (https://www.dagnae.de/), and (5) The Competence Network HIV/AIDS. In each of these institutions, a paper version of the questionnaire was distributed. To participate online, generated flyer strips were also distributed, which included an online link for participation in this study.

The HIBES cohort study itself was also promoted by the consortium of HIV and AIDS specialized physicians (DAGNÄ), the Competence Network HIV/AIDS, the print and online magazines, and the gay social network planetromeo.com. From each of these institutions, the HIBES cohort study was promoted and the online link was presented to their patients, readers, and users.

### 2.4. Definitions

Whereas physical activity represents any bodily movement produced by skeletal muscles that requires energy expenditure, for the purposes of this current study, exercise was used and defined as “planned, structured and repetitive bodily movement done to improve or maintain one or more components of physical fitness” [22]. This study explicitly investigated regular recreational exercise and its characteristic parameters: duration, frequency, and intensity. Specifically, “regular” exercise was defined as performing any kind of exercise at least once a week for at least 3 months [23]. If no exercise was performed regularly, the participants were assigned to the non-exercising group.

The study sample was divided into two main groups: non-exercising and exercising participants. The exercising group was further divided into STE and CTE regarding the amount of types of exercises they were participating in. STE was defined as performing only one type of exercise and CTE was defined as performing two or more types of exercise in parallel throughout the week. “No exercise” was defined as performing no exercise at all during free time. “Volume” was defined as the total minutes of workout durations performed in a week. “Frequency” was defined as the sum of exercise sessions per week. “Intensity” of the exercise was represented by metabolic equivalents METS × minutes per week by including the MET of each performed exercise. The METS × minutes per week were calculated using the International Physical Activity Readiness Questionnaire (IPAQ) guidelines [16].

### 2.5. Data Collection and Measures

Data were collected through a hardcopy or online questionnaire version of the HIBES cohort study in either German or English languages. All questions were assessed through self-report and no blood samples were drawn.

The participants were asked to mark the cardiovascular, metabolic, and psychological diseases listed in the questionnaire following the ICD10 system of disease categorization. Furthermore, HIV-specific or AIDS-defining illnesses were also assessed.

Data on exercise were collected using the questions: (a) do you currently do exercise? (b) Please insert your current exercise activities: types of exercise, session per week, minutes per session. Please mark whether you are performing exercise regularly. Participants were given the chance to enter as many exercise types as they performed.

HRQOL was also assessed through the Medical Outcome Survey-HIV (MOS-HIV) and the Euro-HRQOL 5 dimensions 5 levels (EQ-5D-5L) [24,25,26,27,28]. The MOS-HIV consists of 35 items comprising 11 dimensions, which generate a mental health summary score and a physical health summary score, with higher scores denoting a better HRQOL. For the physical health summary score, the sub-items physical functioning, pain, and role function scale score contribute most strongly; for the mental health summary score, the sub-items mental health, health distress, quality of life, and cognitive function scales contribute most strongly. Vitality, general health, and social functioning contribute to both the mental health summary score and physical health summary score. The MOS-HIV was chosen according to its strong usability for HIV-positive people and can be compared to the short form—36 [29]). The EQ-5D-5L comprises five dimensions of mobility, self-care, usual activities, pain/discomfort, and anxiety/depression in which each dimension has five levels: no problems, slight problems, moderate problems, severe problems, or extreme problems. These values were converted into a single index value using the crosswalk link specified for a German population regarding the EQ-5D-5L manual [27]. In addition, the EQ-5D-5L also comprises the visual analogue scale (VAS). In this scale, the current state of the general subjective health can be cross-marked within a score between 0 and 100 (100 the highest and 0 the lowest subjective general health). The EQ5D-5L was chosen due to its shortness and ease in data evaluation, simplicity, international generalizability, and also construct validity [30].

### 2.6. Data Analysis

To compute the metabolic equivalents (METS) spent in exercise, the Compendium of Physical Activities by Ainsworth et al. was used [31]. The volume, frequency, and intensity were summed for the CTE group.

As noted, center of disease control (CDC) categories were assessed by the HIBES questionnaire. Participants were asked if they currently had an HIV-specific and AIDS-defining illness and were then categorized in stages as A (asymptomatic), B (symptomatic), or C (AIDS-defining) according to the CDC classification (https://www.cdc.gov/mmwr/preview/mmwrhtml/00018871.htm).

### 2.7. Statistical Analysis

To compute the summary statistic, the R package “psych” was utilized. Non-Parametric two-sample Wilcoxon tests were used to compare the central tendencies as the data were not normally distributed. To assess categorical dependence, a chi-square test was used with the stats package in R [32,33]. Correlations were calculated by the same package. Linear regression models were used to calculate the exact relationship between exercise parameters and the HRQOL indexes. ANOVA was used to estimate the variance induced by categorical variables on HRQOL and the exercise variables. For the multiple regression models, the psychological comorbidity was included as a predictor variable coding: 0 = no psychological disorder, 1 = psychological disorder.

R 3.6.1 with dplyr was used for statistical data analysis [33,34] from the R Core Team, Statistical Computing, Vienna, Austria. The package “car” was used to perform the visual residual analysis for ANOVA and multiple linear regressions [35]. To assess influential outliers, the regression results were compared with a robust linear regression calculated with the MASS package [36].

For Pearson correlations, the missing values were not imputed, but complete pairs were correlated. Graphics were created using the package “ggpubr” [37].

## 3. Results

Of the 668 eligible participants, 152 participants needed to be excluded because of no data in the exercise section and 62 participants needed to be excluded because of no data in the HRQOL sections. Then, 454 met the current study inclusion criteria. Most of the respondents were male (92.1%), compared to the smaller proportion of females (7.5%) and other gender (0.4%) participants. The sample mean age was (Median ± Interquartile Range (IQR)) 44 (13) years, a height of 179.5 ± 9 cm, a weight of 77 ± 14, and a body mass index (BMI) of 23.5 (4.3) kg/m^2^ The mean years of HIV infection were 8 (11), the mean CD4 cell count was 635 (307) cells/µL, and the mean initiation of ART was 6 (10) years. Most participants lived in urban areas with a population > 500,000 residents (65.9%). From all included subjects *n* = 72 (15.9%), participants reported one or more cardiovascular diseases, *n* = 101 (22.3%) metabolic disease and *n* = 146 (32.2%) psychological disorders. Only 42 participants received their positive diagnosis before 1990. The year of the first introduction of HIV medication was 1987.

Differences in participant’s distribution between no-exercise and exercise were found according to age, female, and male gender. Participants in the exercise group had a higher level of years of education compared to the non-exercisers (*p* < 0.001). Exercisers seemed to show less comorbidities compared to the no-Exercisers, but only the metabolic comorbidities were significant (*p* < 0.05). No differences in baseline parameters were found for CTE and STE, Table 1.

For HRQOL parameters, the mental health summary score, physical health summary score, index value, and visual analogue scale of the exercise group were all significantly higher (*p* < 0.001) compared to the no-exercising group with small effect sizes (Wilcox r = 0.2 to 0.239). However, no significant differences between CTE and STE were found (*p* > 0.01).

### 3.1. Exercise Compared to Other Covariates

Performing exercise was found to be significantly associated with metabolic comorbidities relative to not exercising (X^2^ = 11.544, *p* = 0.001, Cramer’s V = 0.16). However, exercise status was not associated with cardiovascular and psychological comorbidities, CDC-stage, and ART (*p* > 0.1). STE and CTE were also not associated with those covariates.

#### 3.1.1. HRQOL Parameters

HRQOL (mental health summary score, physical health summary score, index value, and visual analogue scale) were all highly correlated with each other; Pearson r ranged from 0.66 to 0.77.

#### 3.1.2. Covariates Affecting Each Other

Psychological, metabolic and cardiovascular comorbidities, and CDC-stage were all significantly associated with each other (X^2^: *p* < 0.005 for all combinations). ART was also significantly associated with metabolic and cardiovascular comorbidities (X^2^: *p* < 0.05 for both), but not with other covariates.

#### 3.1.3. HRQOL Covariates

Psychological, metabolic, and cardiovascular comorbidities all had a significant negative effect on HRQOL parameters (Wilcox test: *p* = 0.005), whilst psychological comorbidities showed the strongest effect size (Wilcox r = 0.45 to 0.50). The CDC-stage (asymptomatic, symptomatic, and AIDS-defining) also showed a significant effect on HRQOL parameters (Kruskal test: *p* < 0.005, Eta Square [H]: mental health summary score = 0.0639, physical health summary score = 0.136, index value = 0.145, visual analogue scale = 0.0985), Figure 1. However, ART did not have any effect on HRQOL (Wilcox test: *p* > 0.1). Conducting a multifactorial ANOVA with combined exercise, CDC-stage, and psychological disorders, the effect of CDC-stage (symptomatic, AIDS-defining) was no longer significant. Further, 18% of the mental health summary score was explained through psychological disorders, whereas exercise only accounted for 2% (Table A1).

#### 3.1.4. Exercise and its Effects on HRQOL

STE and CTE showed significant large effects on exercise variables (t-test: *p* < 0.005) including number of sessions per week (d = 1.42), minutes per week (d = 1.38), and METS xminutes per week (d = 1.33), Figure 2. Participants performing multiple exercise types showed significantly higher exercise variables than participants performing only one type of exercise: 5 (4) vs. 3 (2) sessions per week (effect size: 0.613, *p* < 0.01), 540 (300) vs. 180 (180) min per week (effect size: 0.657, *p* < 0.01), and 3240 (1950) vs. 1170 (1245) METSxinutes per week (effect size: 0.635, *p* < 0.01) (Table A2).

For all HRQOL scales, the factor “exercise” (levels: no-exercise, STE, CTE) had a significant effect (Two-Way-ANOVA: F(2,450) < 8, *p* < 0.001) when psychological disorder was included as a covariate. For participants with no psychological disorders, a clear difference between non-exercising and exercising participants was evident (Post-hoc t-test, *p* < 0.001, for STE and CTE) for all scales; although, no difference was found between STE and CTE.

For PWH with a psychological disorder, no significant difference was found between the no-exercise and STE groups (*p* > 0.1). However, the mental health summary score (*p* = 0.02), physical health summary score (*p* = 0.03), and visual analogue scale (*p* = 0.01) were higher for the CTE group compared to the no-exercising group. The mental health summary score was also significantly different between the CTE and STE groups (*p* = 0.04).

### 3.2. Linear Regression

A multiple linear regression for all exercising participants with psychological disorders as a covariate was conducted for all HRQOL and exercise variables. No significant association was found between exercise sessions per week and the HRQOL variables. The index value was also not correlated with any exercise variables, Figure 3. No significant interaction between psychological disorder and any exercise predictor was found and therefore excluded from the model. There was no significant relationship between the sessions per week and mental health summary score (beta = 0.03 *p* = 0.51), physical health summary score (beta = 0.00 *p* = 0.88), and visual analogue scale (beta = 0.04 *p* = 0.38).

For mental health summary score, the minutes per week (F(2,273) = 4.907, *p* = 0.028) and METS × minutes per week (F(2,272) = 9.15, *p* = 0.003) showed a significant effect. For the physical health summary score (F(2,271) = 6.148, *p* = 0.014) and the visual analogue scale (F(2,266) = 9.678, *p* = 0.002), only the METS × minutes per week showed a significant effect, Figure 3.

By conducting a simple linear regression for exercising participants with a psychological disorder, a great increase of the slope was observed for the mental health summary score and visual analogue scale: minutes per week and METS × minutes per week; physical health summary score: METS × minutes per week, Table 2. There was no significant relationship between the number of sessions per week and mental health summary score F(1,83) = 0.43, *p* = 0.51, physical health summary score F(1,83) = 0.24, *p* = 0.62, and visual analogue scale F(1,82) = 0.19, *p* = 0.66.

## 4. Discussion

The proportion of EU citizens who never exercise has increased from 39% to 42% [38]. The HIBES study data are compatible with these findings: notably, 36.6% of the participants never participated in any kind of recreational exercise. Moreover, the German cohort study of Stein et al. reported that 61% of participants were actively performing only one type of exercise at the time of the interview [39]; although, the groups comprised participants of mixed HIV status (positive/negative). A quarter (25.5%) of the individuals in Stein et al.’s study participated in weight training or bodybuilding, while 32.2% participated in running, swimming, or cycling. In comparison, the HIBES cohort study provided data of *n* = 288 HIV-positive individuals performing one to three types of exercise.

To date, no published study has investigated the impact of CTE on HRQOL. HIV longitudinal studies examining engagement in exercise or physical activity have mostly evaluated physical activity using self-developed questionnaires or by administering the IPAQ [40,41]. To address this gap, in the current study, the cumulative number of recreational exercise types were assessed (i.e., CTE and STE) compared to individuals who did not exercise as defined by the IPAQ framework. The results revealed differences between exercises (irrespective of whether they did STE or CTE) relative to non-exercisers with HIV. Moreover, randomized controlled trials evaluating exercise have primarily compared a non-exercising group against an exercising condition performing only one type of exercise, e.g., strength, aerobic, concurrent training, yoga, or tai chi as a single type of exercise intervention [42,43,44,45]. Yet, the daily experiences of professional trainers, physiologists, and physiotherapists have observed that people are participating in different types of recreational exercise during the course of a week. In the HIBES study, 38% of the exercising participants were performing more than one type of exercise during a weekly period. This finding represents a more realistic representation of performed exercise types in a German sample of PWH. One of the objectives of the HIBES cohort study was to show a comprehensive picture of CTE and its differences to STE, which has been substantiated by the current findings.

### 4.1. Comparison of Exercise and No-Exercise on HRQOL

Exercisers reported significantly higher HRQOL relative to non-exercisers. Although no significant differences in HRQOL were found between participants in the cumulated exercise group compared to the STE group, a different pattern of findings emerged when psychological disorders were factored into the analyses. Notably, for individuals with a psychological disorder, participants who performed CTE reported higher HRQOL scores compared to individuals who only performed STE per week.

Covariates have a great influence on HRQOL. In this study, psychological disorders and the CDC-stages had a great influence. The presence of a psychological disorder is associated with the possibility of having another comorbidity such as a cardiovascular and/or metabolic disease. The CDC-stages (symptomatic, AIDS-defining) were also associated with psychological disorders.

### 4.2. Association of Exercise and HRQOL

Generally, for all exercising participants, HRQOL scores were positively associated with minutes per week (volume) and METS × minutes per week (intensity). The number of specific sessions per week was not associated with HRQOL.

Exercise variables were associated with psychological disorder status in the exercise group. Considering the heterogeneity of this group, there was still a measurable effect of intensity METS × minutes per week. For participants with psychological disorders, the minutes per week were strongly associated with the mental health summary score, which means that if the participants had a psychological disorder, every minute per week of additional exercise improved their mental health quality of life score. Performing exercise still leads to better HRQOL scores when compared to non-exercising PWH, but with a smaller slope compared to exercisers with a psychological disorder. This pattern was also observed in the physical health summary score and visual analog scale, but not the index value.

High levels of physical activity were associated with a higher quality of life in previous studies (EQ-5D-5L, visual analogue scale) [46]. Moderate and high-volume physical activity has also been associated with higher HRQOL with a strong dose-effect relationship between the intensity of leisure-time physical activity and HRQOL [47]. In the HIBES cohort study, only people performing exercise regularly (at least one session per week) were included and a high frequency, volume, and intensity of exercise was observed in the CTE group.

Notably, the exercise variables in the CTE group were higher when compared to the STE group. Higher training volumes revealed better mental health summary score and physical health summary scores. However, performing CTE was directly associated with an increased HRQOL; although, this was influenced by the exercise variables. These findings can be explained in several ways. First, the variance in CTE and STE was high. Second, although someone in the STE group may train at a high volume (e.g., eight hours of tennis per week), the probability of higher-volume training is more likely in the CTE group. Therefore, it is probable that CTE within its increased likelihood of greater training volume has mediatory characteristics in improving HRQOL.

The best model of the multiple linear regression for all exercisers only predicted 2% of the variance, and for the best model of the simple regression, exercisers predicted 11% of the variance. These results are likely due to the heterogeneous sample. Low HRQOL has been documented to be negatively associated with low employment, education, income level, and comorbidities in PLWH [48,49]. However, psychological comorbidities were included as a covariate in the current analyses. Indeed, results showed that psychological comorbidity had the highest impact on HRQOL-variables. In particular, low income, education, and precarious employment showed a strong positive association with a psychological comorbidity. This justified our decision to evaluate the groups according to psychological comorbidity status.

### 4.3. HRQOL Scales

The scales of the mental health summary score and the physical health summary score in the MOS-HIV usually range from 0–100. For the current study, it is noticeable that the highest score was 67 points. The mental health summary score differed between participants with a comorbidity. More differentiation of the participants was observable in the lower ranges (from 30 to 50) of the HRQOL. Individuals with no comorbidity did not substantially differ from each other. The MOS-HIV scores did not adequately differ between PLW with good HRQOL; although, the differentiation was better for PLH with low to moderate HRQOL. Given these mixed patterns of findings, this suggests that a better HRQOL instrument may be warranted to differentiate participants with a moderate to good HRQOL on lifestyle variables including exercise.

### 4.4. Implications

In the current study, minutes and intensity of exercise predicted HRQOL. However, further research is needed using a longitudinal design to assess the pre-post effects of volume and intensity of exercise, including differentiating between single and cumulated types of exercise on HRQOL, and whether comorbidities also moderate these associations. This study implicates that exercising PLWH show high HRQOL scores when compared to non-exercising PLWH. In addition, this study can be used for practical exercise recommendations regarding the investment of exercise volume, frequency, and intensity to improve HRQOL in PWH with or without the presence of a comorbidity.

### 4.5. Limitations and Strengths

Some limitations are important to consider in interpreting the current findings. First, the HIBES study is a cross-sectional study, so causal inferences cannot be determined. Second, the questions in the exercise section of the HIBES survey were not based on a validated scale. Third, other variables related to both exercise and HRQOL, such as energy intake, nutrition, smoking, drinking, and stress, were not considered in the HIBES study. Fourth, the variance of distribution was very high.

Notwithstanding these limitations, this study differs from other cross-sectional studies by investigating single versus cumulated types of exercise and focusing on the volume, frequency, and intensity of exercise in chronic disease for PLWH. Additionally, the association of exercise and its volume, frequency, and intensity with HRQOL were evaluated, while other studies focused solely on the association between physical activity and HRQOL.

## 5. Conclusions

The findings highlight that on a daily basis, PWH are participating in more than one type of recreational exercise. Not performing any type of exercise is linked to low HRQOL and is also associated with an increased risk of developing a comorbidity. Hence, the findings provide further evidence that exercise is related to better HRQOL in PWH. Moreover, the results show that PWH who have a psychological disorder and who exercise report better HRQOL parameters compared to PWH who exercise but do not have a psychological disorder. The reason for this pattern of findings may in part be due to exercise having a potential synergistic beneficial impact on people’s emotional and physical well-being when struggling with psychological disorders. In contrast, the absence of this comorbidity and performing exercise may maintain people’s psychological resilience.

Furthermore, the outcomes suggest that CTE may serve as mediating interventions to increase exercise volume, frequency, and intensity. This increased exercise variable positively affects HRQOL scores, since the exercise volume is higher with cumulated exercise. Hence, cumulated types of exercise might play an important role in enhancing mental and physical health.

## Figures and Tables

**Figure 1 ijerph-17-05138-f001:**
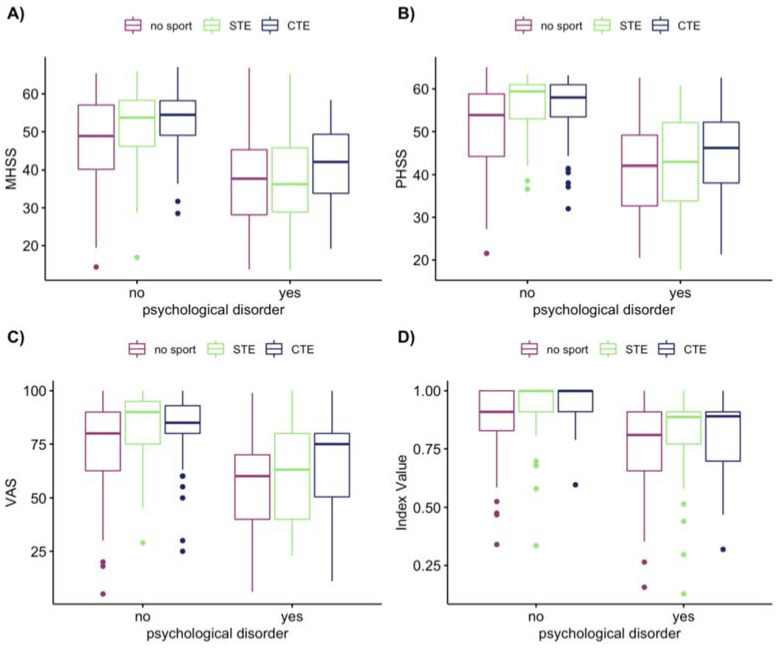
Impact of psychological disorders and exercise on HRQOL. Note: The panels show the difference of the psychological and no psychological covariates between no-exercise, single type of exercise (STE), and cumulated types of exercise (CTE) on the quality of life measures (**A**) MOS-HIV—mental health summary score (MHSS), (**B**) MOS-HIV—physical health summary score (PHSS), (**C**) EQ-5D-5L—visual analogue scale (VAS), (**D**) EQ-5D-5L—Index Value. Medical Outcome Study—HIV (MOS-HIV), Euro-QOL 5 dimensions—5 levels (EQ-5D-5L), HRQOL—Health-related quality of life.

**Figure 2 ijerph-17-05138-f002:**
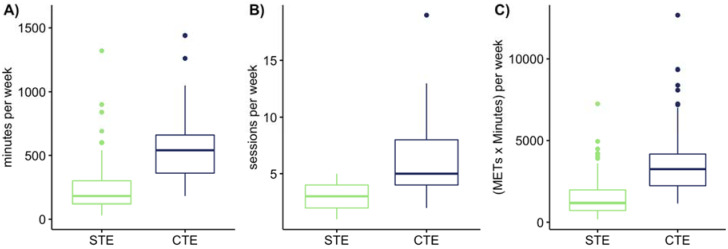
Exercise intensities between single and cumulated types of exercise. Note: The panels show the difference between single type of exercise (STE) and cumulated types of exercise (CTE) on (**A**) minutes per week, (**B**) sessions per week, (**C**) METS × minutes per week.

**Figure 3 ijerph-17-05138-f003:**
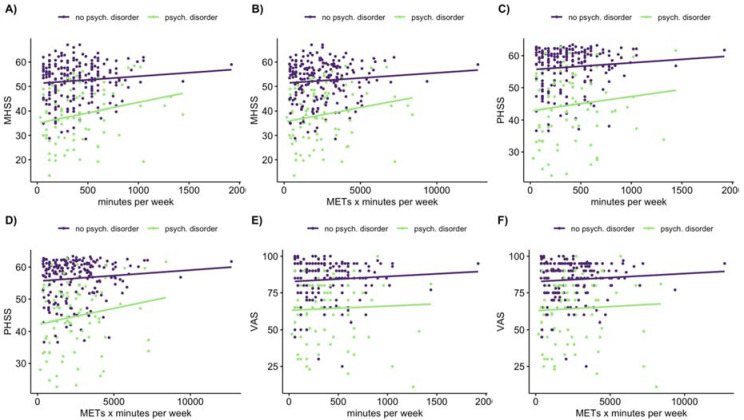
Scatterplot HRQOL and exercise variables. Note: The panels show the difference distributions of participants with or without a psychological disorder on (**A**) mental health summary score (MHSS) and minutes per week, (**B**) mental health summary score and METS × minutes per week, (**C**) physical health summary score (PHSS) and minutes per week, (**D**) physical health summary score and METS × minutes per week, (**E**) visual analogue scale (VAS) and minutes per week, (**F**) visual analogue scale and METS × minutes per week. psych (psychological).

**Table 1 ijerph-17-05138-t001:** Study population characteristics.

Parameter	Non-Exercise	Exercise	CTE	STE
	*n* = 166	*n* = 288	*n* = 141	*n* = 147
Female, *n* (%)	21 (12.7)	13 (4.5) *	5 (3.5)	8 (5.4)
Male, *n* (%)	143 (86.1)	275 (95.5) *	136 (96.5)	139 (94.6)
Other, *n* (%)	2 (1.2)	0 (0)	-	-
Age (yrs.)	46 (12)	43 (13) *	43 (13)	43 (13)
Height (cm)	178 (10.8)	180 (10)	180 (9)	180 (11)
Weight (kg)	75 (17.8)	78 (12)	78 (13)	78 (12)
BMI (kg/m^2^)	23.3 (5)	23.7 (3.8)	23.6 (3.9)	23.7 (3.8)
Residential area				
<100,000 citizens, *n* (%)	31 (18.7)	58 (20.1)	31 (22)	27 (18.4)
100,000–500,000 citizens, *n* (%)	24 (14.5)	42 (14.6)	22 (15.6)	20 (13.6)
>500,000 citizens, *n* (%)	111 (66.9)	188 (65.3)	88 (62.4)	100 (68)
Years with HIV (yrs.)	8 (10)	7 (11)	7 (11)	8 (11)
Year of diagnosis > 1990	151 (90.9)	266 (92.4)	133 (94.3)	133 (90.5)
Year of diagnosis ≤ 1990	15 (9.1)	22 (7.6)	8 (5.7)	14 (9.5)
Years with ART (yrs.)	6 (9)	5 (10)	5 (10)	5 (10)
CD4 cell count (cell·µL^−1^)	600 (330.3)	640 (255.5)	619 (229.5)	650 (336.5)
Cardiovascular disease, *n* (%)	32 (19.3)	40 (13.9)	17 (12.1)	23 (15.6)
Metabolic disease, *n* (%)	52 (31.3)	49 (17) *	25 (17.7)	24 (16.3)
Psychological disorder, *n* (%)	61 (36.7)	85 (29.5)	40 (28.4)	45 (30.6)

Note: Data presented in median and interquartile range (IQR), number of participants (*n*), and percentage (%). Performing cumulated types of exercise (CTE), performing single type of exercise (STE), year (yrs.), anti-retroviral treatment (ART), body mass index (BMI), human immunodeficiency virus (HIV), CD4 cell count expressed in cells per microliter (cell ·µL^−1^). Significant differences at the 0.05 level between No-Exercise and Exercise (*), Chi-Squared-test for comparison of proportions, Wilcox test for comparison of means.

**Table 2 ijerph-17-05138-t002:** Multiple and simple linear regression: exercise and HRQOL.

HRQOL	Value	Estimate (±SE)	Standardized	t	*p*	R^2^	Partial R^2^
**Multiple regression—all exercisers**							
**MHSS**	Intercept	50.46 (±0.97)		52.05	0.000		
	Minutes (slope)	4.86 (±1.89)	0.12	2.57	0.011	0.344	0.023
	Psych. Disorder	−14.00 (±1.17)	−0.58	−12.00	0.000		
**MHSS**	Intercept	50.72 (±0.97)		52.45	0.000		
	MMW (slope)	0.66 (±0.30)	0.11	2.22	0.027	0.340	0.017
	Psych. Disorder	−13.92 (±1.17)	−0.57	−11.91	0.000		
**PHSS**	Intercept	55.32 (±0.84)		66.00	0.000		
	Minutes (slope)	2.91 (±1.65)	0.09	1.77	0.079	0.338	0.011
	Psych. Disorder	−11.93 (±1.00)	−0.58	−11.92	0.000		
**PHSS**	Intercept	55.13 (±0.83)		66.22	0.000		
	MMW (slope)	0.54 (±0.26)	0.10	2.08	0.038	0.341	0.015
	Psych. Disorder	−11.89 (±1.00)	−0.58	−11.90	0.000		
**VAS**	Intercept	81–64 (±1.74)		46.97	0.000		
	Minutes (slope)	6.79 (±3.40)	0.10	2.00	0.047	0.263	0.014
	Psych. Disorder	−20.27 (±2.07)	−0.50	−9.78	0.000		
**VAS**	Intercept	81.57 (±1.73)		47.11	0.000		
	MMW (slope)	1.10 (±0.53)	0.11	2.06	0.040	0.263	0.015
	Psych. Disorder	−20.16 (±2.07)	−0.50	−9.73	0.000		
**Simple regression—exercisers with a psychological disorder**							
**MHSS**	Intercept	35.04 (±1.92)		18.25	0.000		
	Minutes (slope)	9.72 (±3.80)	0.28	2.56	0.012	0.08	
**MHSS**	Intercept	35.44 (±1.94)		18.27	0.000		
	MMW (slope)	1.44 (±0.63)	0.25	2.28	0.026	0.06	
**PHSS**	Intercept	42.39 (±1.94)		21.81	0.000		
	Minutes (slope)	6.56 (±4.05)	0.18	1.62	0.11	0.03	
**PHSS**	Intercept	41.80 (±1.91)		21.91	0.000		
	MMW (slope)	1.32 (±0.65)	0.22	2.03	0.045	0.05	
**VAS**	Intercept	56.29 (±3.90)		14.27	0.000		
	Minutes (slope)	23.75 (±8.47)	0.30	2.80	0.006	0.09	
**VAS**	Intercept	55.30 (±3.89)		14.20	0.000		
	MMW (slope)	4.34 (±1.38)	0.34	3.15	0.002	0.11	

**Note.** Results of multiple regression of exercise parameter and psychological disorder (yes (1)/no (0)) and linear regression just with psychological disordered population. Estimate and Standard Error (SE), Standardized beta coefficient, p-value, coefficient of determination (R^2^), and partial coefficient of determination (partial R^2^). The slope ± SE was multiplied with 1000 (slope × 1000). Health-related quality of life (HRQOL), METS × min per week (MMW), mental health summary score (MHSS), physical health summary score (PHSS), visual analogue scale (VAS), psych (psychological).

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
