# Peer review of "Exercise, Comorbidities, and Health-Related Quality of Life in People Living with HIV: The HIBES Cohort Study"

_ijerph, 2020, doi:10.3390/ijerph17145138_

Round 1

Reviewer 1 Report

-What is hypothesis of study?

-In the abstract: To add more numbers and statistical informations.

Major comments:

-This study must to use only validated scale/questionnaire to determinate the exercise informations. Please, a new evaluation of data, statistical and discussion is crucial.

-More details are required in the Introduction. Please, what is epidemiological informations and comorbidities in HIV patients?

-What is inclusion criteria? Line 173? Please, to add in the methods.

-In the methods: What is linear and multiple regression models were used? Please, to include more informations describing the models.

-In the first paragraph of discussion: Please, delete it. More essential information is described.

-All analyses between: 1) exercise and HRQOL and 2) exercise and covariates psychological must be adjusted by smoking status and alcohol drink.

-The data related to frequency of exercise is poorly explained. 

-Line 300: What is beneficial to use IPAQ or no? Please, to add.

-To add the sample size calculus.

-The conclusion section is poorly wrote. What is hypothesis and main aim of this study?

Reviewer 2 Report

The topic of the paper is interesting, and methodology is adequate. However, some definition of concepts regarding the exercise level assessed by the short IPAQ form. It should be remembered that in training the concepts of volume, intensity, duration and frequency are specifically defined, where “duration” and “intensity” are referred to a single workout, while “volume” is the sum total of workout durations for a period of time such as a week. The short IPAQ form is used to assess the physical activity (PA) level and it is structured to provide separate scores on walking, moderate-intensity and vigorous-intensity activity. Computation of the total score for the short form requires summation of the duration (in minutes) and frequency (days) of walking, moderate-intensity and vigorous-intensity activities, recalled over the past seven days. MET-minute scores are quantified by multiplying the MET score of an activity by the minutes performed. Therefore, the assessed parameters are: IPAQ Total PA minutes (min/week), IPAQ Total MET-minutes (min/week), IPAQ Walking MET-minutes (min/week), IPAQ Moderate MET-minutes (min/week), IPAQ Vigorous MET-minutes (min/week). It should be considered that the parameter MET*minutes/week include in itself both the “intensity” and the “volume” so that is referred as physical activity level. The Authors should consider these concepts and made consistent thought the text the definitions of these parameters.

Reviewer 3 Report

Cross-Sectional Associations of Single and Cumulated Types of Recreational Exercise, Comorbidities, and Health-Related Quality of Life in People Living with HIV: The HIBES Cohort Study.

Thank you for the opportunity to review this manuscript. This is a well-written manuscript that presents interesting results. The authors have conducted an interesting study with novel results. They have made a correct statistical analysis and have explored the data in depth showing consistent conclusions.

I suggest minor changes below:

Title: the authors should try to shorten the title.

Abstract: the authors must explicitly include the objective of the study.

  1. Introduction:

Although the introduction presents the main points, I think they should be more explicit about the research problem.

  1. Materials and Methods

Regarding the sample, I think the time of diagnosis of the disease should have been assessed. Because, I think it's a moderating factor.

In line 143 has not included the reference.

  1. Results

The authors have made a good presentation of the results obtained in the statistical analysis.

  1. Discussion and conclusion

The conclusions are, according to the debate, well established and consistent with the initial objectives.

Round 2

Reviewer 1 Report

Accept